# 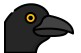 CROW: Benchmarking Commonsense Reasoning in Real-World Tasks

**Mete Ismayilzada[1], Debjit Paul[1*], Syrielle Montariol[1*], Mor Geva[2], Antoine Bosselut[1]**

[1]EPFL, Switzerland     [2]Google DeepMind
mahammad.ismayilzada@epfl.ch

## Abstract

Recent efforts in natural language processing (NLP) commonsense reasoning research have yielded a considerable number of new datasets and benchmarks. However, most of these datasets formulate commonsense reasoning challenges in artificial scenarios that are not reflective of the tasks which real-world NLP systems are designed to solve. In this work, we present CROW, a manually-curated, multi-task benchmark that evaluates the ability of models to apply commonsense reasoning in the context of six real-world NLP tasks. CROW is constructed using a multi-stage data collection pipeline that rewrites examples from existing datasets using commonsense-violating perturbations. We use CROWto study how NLP systems perform across different dimensions of commonsense knowledge, such as physical, temporal, and social reasoning. We find a significant performance gap when NLP systems are evaluated on CROWcompared to humans, showcasing that commonsense reasoning is far from being solved in real-world task settings. We make our dataset and leaderboard available to the research community.[1]

## Introduction

Commonsense reasoning is a long-standing challenge in artificial intelligence (AI) and NLP (McCarthy 1960; Winograd 1974; Davis and Marcus 2015; Choi 2022), resulting in a large number of datasets and benchmarks designed to evaluate how AI systems reason in commonsense scenarios described in natural language (Davis 2023). Recently, large language models (LLMs) such as GPT-3 (Brown et al. 2020) and PaLM (Chowdhery et al. 2022), have demonstrated near-human performance on many of these benchmarks (Lourie et al. 2021). However, these models can still be brittle in practical deployments, raising questions about how reliably these commonsense benchmarks truly evaluate the commonsense reasoning abilities of models.

Part of this issue stems from the practice that most commonsense datasets are designed to evaluate reasoning in artificial task settings that are not reflective of the real-world use cases in which NLP systems are deployed. In real-world settings, one almost never directly observes a test of commonsense knowledge in isolation. In this paper, we argue instead that *commonsense reasoning benchmarks should evaluate commonsense reasoning in the tasks in which these abilities are required.* The necessity of commonsense to solve real-world tasks has been extensively argued since the early stages of AI, notably by Bar-Hillel (1960) in the context of machine translation. However, despite these early arguments, only recently was there an attempt to construct a commonsense reasoning dataset for machine translation (He et al. 2020), an effort which concluded that the commonsense reasoning abilities of modern models were still in their infancy when applied in real NLP tasks.

In this work, we build on these original ideas and introduce **CROW**: a **C**ommonsense **R**eas**o**ning Benchmark for Real-**W**orld Tasks, a benchmark containing high-quality datasets for six real-world NLP tasks: machine translation (MT), open-domain dialogue (DG), dialogue summarization (DS), intent detection (ID), stance classification (SC), and safety detection (SD). We design a flexible multi-stage data collection pipeline to build CROW. Inspired by Winograd schemas (Levesque, Davis, and Morgenstern 2011), we apply commonsense-based minimal perturbations on examples from existing datasets for each task. Then, we crowdsource collections of potential target references, each grounded to a particular commonsense violation with respect to the original context. We categorize these commonsense violations across six dimensions — temporal, causal, attribution, comparison, physical, social — ensuring a diverse breakdown of commonsense reasoning types.

Our empirical study across 13 state-of-the-art (SoTA) systems (including GPT-4) shows that CROW is a challenging commonsense reasoning testbed, with the highest performing model scoring ∼18% lower than humans on individual examples and ∼37% lower on our more restrictive metric that evaluates situational robustness. Consequently, we provide CROW to the community as the first commonsense benchmark specifically formed to test commonsense knowledge and reasoning abilities in the same contexts as real-world deployments of NLP systems.

## Benchmark

In order to construct CROW, we take the simple and effective idea of commonsense-violating minimal perturbations used in the earlier benchmarks to generate Winograd-

---

*These authors contributed equally.

[1]https://github.com/mismayil/crow

| Models | MT | | | | DG | DS | SC | SD | ID | CROW Score (-MT) | CROW Score |
|---|---|---|---|---|---|---|---|---|---|---|---|
| | Zh-En | En-Fr | En-De | En-Ru | | | | | | | |
| LLaMA-33B* | 50.5 / 1.2 | – | – | – | 50.5 / 2.6 | 48.2 / 7.8 | 57.1 / 0.0 | 44.1 / 4.1 | 42.4 / 1.2 | 48.5 / 3.1 | 48.8 / 2.8 |
| Flan-T5-11B* | 45.5 / 10.1 | – | – | – | 70.4 / 42.0 | 66.9 / 33.1 | 76.5 / 51.6 | 83.8 / 34.9 | **84.3 / 57.7** | 76.4 / 43.9 | 71.2 / 38.2 |
| PaLM-1-540B | 52.7 / 5.7 | 50.2 / 0.4 | 50.0 / 0.0 | 50.0 / 0.0 | 63.4 / 24.7 | 61.2 / 20.2 | 51.3 / 19.1 | 49.5 / 7.7 | 70.4 / 32.3 | 59.2 / 20.8 | 55.4 / 12.2 |
| GPT-3.5 | 66.6 / 38.7 | 50.1 / 18.2 | 50.6 / 18.0 | 48.9 / 13.2 | 67.6 / 36.5 | 68.7 / 31.9 | 67.7 / 36.0 | 85.6 / 40.0 | 76.4 / 41.7 | 73.2 / 37.2 | 64.7 / 30.5 |
| GPT-4 | **75.9 / 57.9** | 54.5 / 21.5 | 54.4 / 20.5 | 54.1 / 19.7 | **72.4 / 46.5** | **89.6 / 75.3** | 79.6 / 54.7 | **89.7 / 51.9** | 84.0 / 57.2 | **83.1 / 57.1** | **72.7** / 45.0 |
| GPT-4-CoT | 71.6 / 52.2 | **64.7 / 42.6** | **57.1 / 34.2** | **57.3 / 30.0** | 55.3 / 22.8 | 88.6 / 70.6 | **84.3 / 60.7** | 87.8 / 47.3 | 84.0 / 57.0 | 80.0 / 51.7 | 72.3 / **46.4** |
| Human | 87.9 / 78.0 | 83.0 / 82.9 | 89.9 / 82.0 | 89.9 / 86.0 | 87.0 / 86.9 | 98.9 / 96.4 | 88.1 / 69.6 | 97.8 / 93.9 | 93.9 / 80.7 | 93.1 / 85.5 | 90.7 / 84.0 |

Table 1: **Macro-F1 / Situational Accuracy** (*i.e.*, results aggregated per *context* instead of per *sample*) for all examined models across CRoW tasks. The performance of the highest scoring model is **bolded** for each task. Models noted with * are open-source. Since they are trained only on English data, we don't evaluate them on MT tasks except the one translating to English. Due to the cost of expert evaluation, our **Human** study is only evaluated on 100 instances per task.

style schemas (Davis 2023) and apply it in real-world tasks. We employ crowdsourcing to generate Winograd-style perturbed examples, but instead of asking crowdworkers to perturb the given sentences directly, we design a data collection pipeline that breaks down the schema construction into two independent stages: **Commonsense Knowledge Annotation** and **Winograd-style Schema Generation**, each of which is followed by a complementary validation stage. For a given task example, we define the *context* as the unchanged part of the example and the *target* as the candidate for commonsense-based minimal perturbation. In the first stage of our pipeline, we explicitly annotate implicit commonsense knowledge connecting a *context* and a *target* in real-world task datasets. In the second stage, we present workers with the *context*, the *target*, and the associated commonsense knowledge from the previous stage, and ask them to rewrite the *target* such that it satisfies the following four conditions. The new *target* must (1) minimally differ from the original target (*i.e.*, by edit distance of at most five words), (2) directly violate the given commonsense knowledge, (3) be an incorrect answer for the given context, and (4) be contextually relevant.

We use Amazon Mechanical Turk as a crowdsourcing platform. Our final benchmark contains ∼5K unique contexts with ∼500 unique contexts per task (on average) and ∼16K examples (*i.e.*, context-target pairs) in total. Following is an example from our benchmark for Intent Detection task: **Headline (context):** *Remote glaciers in China melting at 'shocking' pace, risking water shortages*, **True Intent (target):** *Climate change is real and is showing its effects*, **Knowledge:** *water shortages IsA effect*, **False Intent (target):** *Climate change is real and is showing its causes*.

## Experiments and Results

All tasks in CRoW are treated as binary classification tasks. Given a *context*, a model must predict whether a provided *target* is a suitable response for the corresponding real-world task. For instance, in machine translation, given an English sentence and a translated sentence in French, the model must predict whether the translation is valid.

**Evaluation Metrics.** We evaluate models on CRoW using two scores: **Macro-F1** (valid or invalid *targets*), and **Situational Accuracy**, a stringent metric that reports whether the model correctly identifies the validity (or invalidity) of

all *targets* for a given *context* (similar to Storks and Chai 2021's strict coherence score). A single mistake on any *target* results in a score of 0 for that context. We design this metric to account for the fact that robust commonsense reasoning would provide the model with a full situational understanding of the *context*. The CRoW score is computed as a macro-average of the task scores. We evaluate the human performance on each task of the benchmark using two expert annotators who each evaluate 100 random task samples.

**Results** We evaluate a series of language models that are diverse in terms of scale, training, and data. Table 1 reports results for a few of the latest LLMs across all tasks. For all results, we refer the reader to our full paper.[2] In general, we observe that models vary in their ability to correctly identify the correct responses in the tasks. As expected, GPT-4 outperforms most other models. Even among stronger models, though, while performance is higher for individual examples (as measured by Macro-F1), the situational accuracy is significantly lower, often below 50%. This gap suggests that these models are not robust and fail to grasp a full situational understanding of the contexts with which they are presented (even as they may correctly classify some individual cases). In contrast, humans tend to perform well on both metrics. We also perform an analysis on the effect of oracle commonsense knowledge and show that models can leverage it to improve their performance. Our qualitative analysis on the other hand reveals some patterns (e.g. less plausible reasoning due to imagining sarcastic scenarios) underlying the surprisingly weaker performance of GPT-4 with chain-of-thought reasoning with respect to GPT-4 with standard prompting (especially, on Dialogue task).

## Conclusion

In this work, we propose CRoW, a multi-task commonsense reasoning benchmark consisting of six real-world tasks. To construct our benchmark, we design a data collection pipeline to systematically crowdsource Winograd-style schemas based on commonsense-violating minimal perturbations. Our evaluation of recent LLMs on our benchmark shows that the performance of state-of-the-art models still falls far below human performance with respect to commonsense reasoning in real-world contexts.

---

[2]https://aclanthology.org/2023.emnlp-main.607

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
