# OpenReview forum: "CRoW: Benchmarking Commonsense Reasoning in Real-World Tasks"
_AAAI.org/2024/Workshop/NuCLeaR — NuCLeaR 2024_

### Official Review · Reviewer_meQp · 2023-12-06
**Interesting extended abstract of strong published benchmark**

**Rating:** 8
**Confidence:** 4

**Review:**

CRoW is an interesting benchmark for commonsense reasoning published at EMNLP (I could not find the proceedings to check). Evaluating the capabilities of commonsense knowledge seems relevant to the topic of the workshop. The extended abstract gives a good overview of the paper, and the results are interesting and useful for this community.

The only thing I'm missing is an example of (one of the) tasks to get a better idea of what the benchmark looks like.

---

### Official Review · Reviewer_rDnp · 2023-12-08

**Rating:** 8
**Confidence:** 3

**Review:**

Strengths:
1. The table in the paper very clearly shows the test results on the benchmark.
2. As per shown by the results, the benchmark provides a reasonable evaluation of the capability of LLMs on commensense reasoning.
3. The designed workflow is reasonable.
4. The quality of the data is high, with a clearly defined task and human benchmark.

Weakness:
1. The length of the paper is relatively short, as there could be more imperial analysis done to further demonstrate the characteristics of the benchmark.
2. There could be more in-depth tasks designed based on the benchmark besides a binary task to further demonstrate the reasoning capabilities of the LLMs.

---

### Official Review · Reviewer_FhBs · 2023-12-08

**Rating:** 5
**Confidence:** 4

**Review:**

This paper presents a new commonsense reasoning benchmark called CROW that is based on real-world tasks. The CROW is a manually curated, multi-task benchmark, and it tests the commonsense reasoning ability of a model across six real-world NLP tasks. This benchmark has around 16k examples. The authors have benchmarked the dataset with sota LLMs and empirically showcased it is very challenging.

Strengths:
* This real-world commonsense benchmark will help judge the CS reasoning capability of an LLM.
* Evaluation results showcase the difficulty of this benchmark.
* This is human annotated benchmark.

Weaknesses:
* No examples of curated data have been given, also hard to understand the uniqueness of the data generation approach.
* CS violation across 6 dimensions would be very helpful, however, no statistics have been shown on that.
* Also, good to see how this dataset will help an LLM to improve the CS reasoning performance. Maybe through Finetunining.

---

### Decision · Program_Chairs · 2023-12-11

Accept